# Long-Term Results after Placing Dental Implants in Patients with Papillon-Lefèvre Syndrome: Results 2.5–20 Years after Implant Insertion

**DOI:** 10.3390/jcm11092438

**Published:** 2022-04-26

**Authors:** Katrin Nickles, Mischa Krebs, Beate Schacher, Hari Petsos, Peter Eickholz

**Affiliations:** 1Department of Periodontology, Center for Dentistry and Oral Medicine (Carolinum), Johann Wolfgang Goethe-University Frankfurt am Main, 60596 Frankfurt, Germany; schacher@em.uni-frankfurt.de (B.S.); petsos@med.uni-frankfurt.de (H.P.); eickholz@med.uni-frankfurt.de (P.E.); 2Department of Oral Surgery and Implantology, Center for Dentistry and Oral Medicine (Carolinum), Johann Wolfgang Goethe-University Frankfurt am Main, 60596 Frankfurt, Germany; mischa@dr-krebs.net

**Keywords:** dental implants, Papillon-Lefèvre syndrome, peri-implantitis, periodontitis, long-term results

## Abstract

Aim: A retrospective evaluation of patients with Papillon-Lefèvre syndrome (PLS) treated with dental implants to identify factors that may influence treatment outcomes. Methods: All PLS patients with dental implants currently registered at the Department of Periodontology, Goethe-University Frankfurt (20–38 years; mean: 29.6 years), were recruited. Five patients from three families (two pairs of siblings) with a total of 48 dental implants (inserted in different dental institutions) were included with a follow-up time of 2.5–20 years (mean: 10.4 years). Results: Implant failure occurred in three patients (at least 15 implants). Nearly all patients demonstrated peri-implantitis in more or less advanced stages; 60% of patients demonstrated bone loss ≥50% around the implants. Two patients did not follow any supportive therapy. Conclusions: Implants in PLS patients who did not follow any maintenance programme had a high risk of peri-implantitis and implant loss.

## 1. Introduction

Papillon-Lefèvre syndrome (PLS) is an infrequent genetic disorder characterised by palmoplantar hyperkeratosis combined with rapidly progressive severe periodontitis affecting both the deciduous and permanent dentitions [1]. The prevalence of PLS is 1–4 per million [2] with no sex or race predominance and is inherited as an autosomal-recessive trait. A loss-of-function mutation affecting the cathepsin C gene (CTSC) on chromosome 11q14.1-q14.3 has traditionally been related to the disorder [3,4], its main functions being protein degradation and proenzyme activation [5].

As periodontal therapy often fails in PLS patients [6,7,8], they typically lose their teeth early in life and eventually become edentulous with significant ridge resorption. For an increasing number of cases of PLS patients, it is reported that periodontitis may be arrested even in the long term. In those cases, therapy consists of treatment of the infection with the extraction of severely diseased teeth, combined mechanical and antibiotic periodontal treatment, oral hygiene instructions, intensive maintenance therapy, and microbiological monitoring [9,10,11].

However, PLS patients who have lost many or all teeth need prosthetic rehabilitation. Over the last few years, dental implants have become a common treatment alternative to replace missing teeth. The use of dental implants in young patients with rapidly progessing periodontitis (1999 classification: aggressive periodontitis; 2018 classification: periodontitis grade C) has already been reported [12,13]. Patients with periodontitis grade C (GAgP) had a five times greater risk of implant failure, a three times larger risk of mucositis, and 14 times higher risk of peri-implantitis [13]. Swierkot et al. concluded that patients with treated periodontitis grade C (GAgP) are more susceptible to mucositis and peri-implantitis and experience lower implant survival and success rates than periodontally healthy individuals [13].

Based on the fact that PLS is a rare disease, there are limited studies (mainly case reports) with small numbers of patients assessed in the literature. To the best of our knowledge, just eight articles report outcomes of dental implants in PLS patients [11,14,15,16,17,18,19,20]. Furthermore, long-term results are only occasionally reported.

Therefore, the aim of the present retrospective study was to analyse (long-term) outcomes of dental implants in five PLS patients and to identify factors that may influence treatment outcomes. As is presently best known, this is the largest group of PLS patients treated with dental implants reported thus far.

## 2. Materials and Methods

### Patients and Data Collection

We studied all PLS patients with dental implants registered at the Department of Periodontology, Center for Dentistry and Oral Medicine, Johann Wolfgang Goethe-University Frankfurt (20–38 years; mean: 29.6 years). In all patients, the diagnosis of PLS was based on the clinical findings during the initial examination and confirmed by detecting mutations in the cathepsin C gene by analysing blood samples. The study was registered by the Institutional Review Board for Human Studies of the Medical Faculty Goethe-University under the number 31/05 in 2005.

Five patients (four female) were included. They belonged to three families and included two pairs of siblings. Implant therapy in two patients was performed exclusively at the Department of Oral Surgery and Implantology in Frankfurt, and in the three other patients, it was carried out at external dental clinics or by local dentists or oral surgeons. The initial periodontal therapy in Patients 1 and 2 had been described previously [21] as well as the periodontal development of Patients 1, 2, 3, and 4 [11]. All data were collected from the documents in the patients’ files. The patients included in our study are listed in Table 1.

## 3. Results

### 3.1. Implant Therapy

Patients 1 and 2 were treated at the Department of Oral Surgery and Implantology, Center for Dentistry and Oral Medicine (Carolinum), Johann Wolfgang Goethe-University Frankfurt am Main with the same surgical technique (for details, see Table 1). Despite being provided with extensive information about the importance of supportive implant therapy (SIT), Patient 1 participated in just one maintenance visit. Supportive implant therapy (SIT) consisted of oral hygiene instructions, professional implant cleaning (professional mechanical plaque removal [PMPR] and stain removal), and—bi-annually—a comprehensive periodontal/peri-implant examination. In this way, a rapid intervention was possible whenever needed. In probing pocket depths of 4 mm that showed bleeding on probing (BOP) and/or pockets that were deeper than 4 mm, subgingival/mucosal instrumentation (SI) with titan curettes and/or air-polishing with glycine powder, as well as instillation of 1% chlorhexidine gel, was performed. In the course of every maintenance visit, oral hygiene indices were assessed.

In patient 1, peri-implant mucositis and in two implants, peri-implantitis lesions could be detected already. In Patient 2, compliance improved over time. However, the patient participated in SIT only once a year. Peri-implant mucositis could also be clearly diagnosed in Patient 2 (Table 1).

In Patients 3 (see Figure 1), 4, and 5, implants were primarily inserted years before in different private practices (for details, see Table 1).

### 3.2. Clinical, Microbiological and Radiological Findings

At six sites per implant (mesiobuccal, midbuccal, distobuccal, distooral, midoral, mesiooral), probing pocket depths (PPD) were measured using a manual rigid periodontal probe (PCP UNC15, Hu-Friedy, Chicago, IL, USA) to the nearest millimetre. Bleeding on probing (BOP) was recorded 30 s after probing. Suppuration was documented for each implant (see Table 2). Three patients were already exhibiting PPD ≥ 7 mm. All patients exhibited high BOP scores (>20%) except for Patient 4, who took systemic antibiotics at the time of scoring.

Except for Patient 4, in each patient that was treated with systemic antibiotics, a microbiological examination was performed with sterile paper points from the deepest pocket of each quadrant. For analysis, a commercially available real-time PCR (Meridol Paro Diagnostik Test, Carpegen, Münster, Germany) for the quantitative determination of six periodontal pathogens (*Aggregatibacter actinomycetemcomitans*, *Porphyromonas gingivalis*, *Tannerella forsythia*, *Treponema denticola*, *Prevotella intermedia*, and *Fusobacterium nucleatum*) was employed. None of the patients showed subgingival presence of the periodontal key pathogen, *Aggregatibacter actinomycetemcomitans* (see Table 2).

Panoramic radiographs were either performed on the day of investigation in the university hospital or earlier by private practices and collected at the following appointment. The proportional bone loss was determined by use of a Schei-ruler [22] on the mesial and distal aspect of each implant (distance of implant shoulder to implant apex) and was classified into three categories (Bone loss 0− < 25%, 25− < 50%, ≥50%) (see Table 3). Three (60%) patients demonstrated bone loss ≥50% around the implants.

## 4. Discussion

PLS is a rare genetic disease characterised by hyperkeratosis of the palms and soles. It also manifests in a rapidly progressive, severe periodontitis that leads to premature loss of the primary and secondary teeth if not treated early and consequently. A mutation affecting the CTSC gene on chromosome 11q14.1-q14.3 has been associated with the disorder [3]. The cathepsin C enzyme is expressed by epithelial and immune cells and mainly acts as a key enzyme in the activation of granule serine proteases, e.g., elastase. Several studies have studied the pathogenesis of periodontitis in PLS patients. Compromised neutrophil function, including phagocytosis, chemotaxis, and bacterial killing [23], as well as severely depressed natural killer cell cytotoxicity, have been described in patients with PLS [24,25]. Hence, it is plausible that patients with PLS are also very likely to develop disease around dental implants. For this reason, the use of dental implants in patients with severe forms of periodontitis secondary to systemic disorders was not a treatment option for a long time. As a result of young patients having a need for oral rehabilitation that would otherwise not be treated with fixed prosthetics, the question arises of whether dental implants could also elicit success in PLS patients. Swierkot et al. assessed the prevalence of peri-implant mucositis, peri-implantitis, implant success, and survival in patients with GAgP/periodontitis grade C and in periodontally healthy individuals [13]. They reported implant survival rates of 100% in periodontally healthy individuals versus 96% in patients with GAgP/periodontitis grade C. Further, the implant success rate was 33% in GAgP/periodontitis grade C patients and 50% in periodontally healthy patients. The implant success rate was defined by the following parameters: (1) no implant movement; (2) no discomfort (pain, foreign body sensation, paresthesia); (3) PD ≤ 5 mm without BOP; (4) no continuous radiologic translucency; and (5) annual peri-implant bone loss ≤ 0.2 mm 1 year after insertion of the superstructure. Implants that failed to meet ≥1 criteria were considered a failure. In the GAgP/periodontitis grade C group, peri-implant mucositis could be detected in 56% and peri-implantitis in 26% of the implants. In the periodontally healthy group, 40% of the implants exhibited mucositis and 10% peri-implantitis. In addition, GAgP/periodontitis grade C patients demonstrated a five times greater risk of implant failure, a three times higher risk of mucositis, and 14 times more obvious risk of developing peri-implantitis. Ultimately, the authors contended that patients with treated GAgP/periodontitis grade C are more susceptible to mucositis and peri-implantitis and had lower success rates and implant survival [13]. Actually, to the best of our knowledge, just eight articles have reported the outcomes of dental implants in PLS patients [11,14,15,16,17,18,19,20]. The exception is Nickles et al. [11]—all others only described the outcome in a single PLS patient with a follow-up period of up to 4.5 years. At first sight, the results are positive. Here, we present data from five PLS patients treated with dental implants. In two patients, dental implants were inserted at the Department of Oral Surgery and Implantology at the Johann Wolfgang Goethe-University Frankfurt am Main with state-of-the-art techniques. In one patient, peri-implantitis was already documented after 2.5 years (see Figure 2). Non-compliance to SIT seemed to be the most probable cause of peri-implant destruction in this patient.

All patients treated with dental implants many years earlier (approximately 20 years in Patients 3, 4, and 5) exhibited advanced bone loss around the implants and suffered substantial implant loss. What could be the reasons for these failures? In two patients, Brånemark implants were utilised. The Brånemark system is a well-documented implant system—Ross-Jansåker et al. evaluated the long-term results of implant therapy with implant loss as the outcome variable. In 294 patients, Brånemark implants were inserted between 1988 and 1992 in Kristianstad (Sweden). One and five years after the placement of the suprastructure, the patients were scheduled to the clinic. Between 2000 and 2002 (9–14 years after implant insertion), the patients again underwent a clinical and radiographic examination. In total, 218 patients treated with 1057 implants were assessed and the overall implant survival rate was 95.7%. A significant connection could be noticed between implant loss and periodontal bone loss of the residual teeth. Overall, it appeared that a history of periodontitis was related to implant loss [26].

Documentation of long-term results with the IMZ^®^ system is rare. Haas et al. presented a cumulative survival rate of 83.2% after 100 months in a study with 1920 IMZ^®^ implants. The results demonstrated a statistically significant lower cumulative survival rate of maxillary (37.9%) versus mandibular implants (90.4%) [27]. In contrast to these findings, Willer et al. documented similar survival rates for upper and lower jaw implants in a prospective observation of 1250 IMZ dental implants. The cumulative survival rate after 10 years (82.4%) was very similar to the findings of Haas et al. [28] The cumulative survival rates associated with the IMZ system (83.2%/82.4%) seemed to be lower than those presented by a working group from Frankfurt University [29] with the Ankylos^®^ Implant system with a survival rate of 93.3% after 204 months (17 years). Whether the implant system utilised in our three PLS cases had any influence on the outcomes remains questionable. 

In 2014, a Cochrane review was published by Esposito et al. [30] concerning the success rates of different types of dental implants. Based on the available results of randomised clinical trials (RCTs), the authors felt there was limited evidence demonstrating that implants with relatively smooth (turned) surfaces were less prone to bone loss because of chronic infection (peri-implantitis) than those with rougher surfaces. On the other hand, there was no evidence indicating that any particular type of dental implant had superior long-term success. These results were based on a small number of RCTs, often at high risk of bias, with few participants and relatively short follow-up periods. More RCTs should be conducted with a follow-up of at least 5 years that also ascertain the inclusion of a sufficient number of patients in order to detect a true difference [30].

As tooth loss in PLS patients is usually accompanied by severe loss of alveolar bone structures in the mandible as well as in the maxilla, bone-grafting methods seem to be particularly necessary with respect to the maxilla. In the mandible, an inter-foraminal implant placement appears to be possible even in cases with severe resorption when short- and narrow-diameter implants are used. These implants seem to be similarly successful to longer implants [29,31].

In the maxilla, depending on the amount of bone loss and the desired form of prosthetic reconstructions, vertical grafting with autologous bone transplants and sinus grafting are apparently possible options for implant-retained reconstructions. Sinus graftings seem to be equally successful when performed with bovine substitutes or autologous materials [32,33,34,35,36]. Iliac bone was employed in at least one of the three PLS patients here with a follow-up period of approximately 20 years. Fretwurst et al. [37] examined the long-term results after onlay grafting with iliac bone. The authors could demonstrate that in patients with atrophic jaws, an adequately long-term reconstruction could be achieved with iliac onlay grafting in combination with dental implants.

Another issue common in the literature was the time at which the implants were placed. In Patients 3, 4, and 5, the first implants were placed by and next to remaining teeth, meaning there was no edentulous period for these patients.

The most important reason for peri-implant disease, however, was the lack of any professional supportive periodontal/peri-implant therapy. Rocuzzo et al. [38] compared the long-term outcomes of implants placed in patients treated for periodontitis (periodontally-compromised patients; PCPs) and in periodontally healthy patients (PHP) in relation to the adherence of SPT. It was observed that patients with a history of periodontitis had a lower survival rate and a statistically significantly higher number of sites with peri-implant bone loss. Furthermore, PCPs that did not faithfully adhere to SPT exhibited a higher implant failure rate. This underlines the value of SPT for enhancing the long-term outcomes of implant therapy, particularly in subjects affected by periodontitis, in order to control reinfection and limit biological complications [38].

Although in Patients 3 and 4, professional dental/implant cleanings (professional mechanical plaque removal [PMPR] and stain removal) were performed every three months, no measurement of probing depths, assessment of BOP/suppuration, or subgingival/-mucosal cleaning took place. Instead, systemic amoxicillin + clavulanic acid and metronidazole were prescribed for seven days twice a year, though no subgingival cleaning was performed concordantly. Altogether, no professional supportive therapy was conducted on these patients. In Patient 5, no supportive therapy took place at all.

Fazele et al. [39] assessed the success of dental implant placement in PLS patients in a systematic review: the authors studied 15 cases with 136 dental implants and they concluded that dental implants may be a viable treatment option for PLS patients and implantation can help preserve alveolar bone if the patients’ immunological and growing conditions are well-considered and proper oral hygiene and compliance with the maintenance program are continued. Nevertheless, in 3 patients, 20 implants failed.

Some PLS patients receive systemic retinoid medication. In the literature, supposedly positive effects have been reported for a systemic medication with oral retinoids [40,41], but also not for others [42]. One of our patients (patient 2) is receiving a systemic retinoid (acitretin) for several years now—despite this, she has lost all of her teeth.

## 5. Conclusions

A history of periodontal disease is a risk factor for peri-implant disease in general and PLS periodontitis in particular. Thus, PLS patients are high-risk patients with regard to peri-implant disease. We report 5 PLS patients losing all teeth and being treated with dental implants. Only one patient receives proper (oral hygiene indices, PPD charting, PMPR, SI) SPT on a yearly basis. Patient 1 received just one proper SPT, Patients 3 and 4 received twice a year systemic antibiotics but only supramucosal PMPR, Patient 5 did not receive any maintenance treatment whatsoever. Thus, two main factors seem to drive bone loss: (1) time (which is trivial) and (2) lack of SPT (which is particularly evident in comparison to Patients 1 and 2).

In light of what we have presented in this work, it remains debatable whether implants should be used in patients with PLS-associated periodontitis. The impaired immune system in PLS patients represents a risk factor that cannot be controlled. These patients have to be classified as high-risk patients and informed of their circumstances. Many of these patients lose their teeth very early, yielding orthodontic and physiognomic, and, hence, psychosocial consequences. Implants often represent the only opportunity to insert fixed or at least stable prostheses in these patients. PLS patients—along with their treating dentist—should be aware of the risks associated with not complying with the prescribed regimen of supportive care, i.e., peri-implantitis and implant loss. Therefore it is of crucial importance that PLS patients are informed about the importance of supportive therapy.

The authors are aware of the fact that the number of patients included in the study was very small. Nevertheless, the manuscript clearly provides the prevalence, i.e., 1 to 4 under 1 million population. The authors are also aware of the fact that the treatment modalities (implant types, various bone grafts, etc.) are very heterogeneous and hard to compare.

In spite of everything, the present study represents the largest on implant treatment in PLS patients so far.

## Figures and Tables

**Figure 1 jcm-11-02438-f001:**
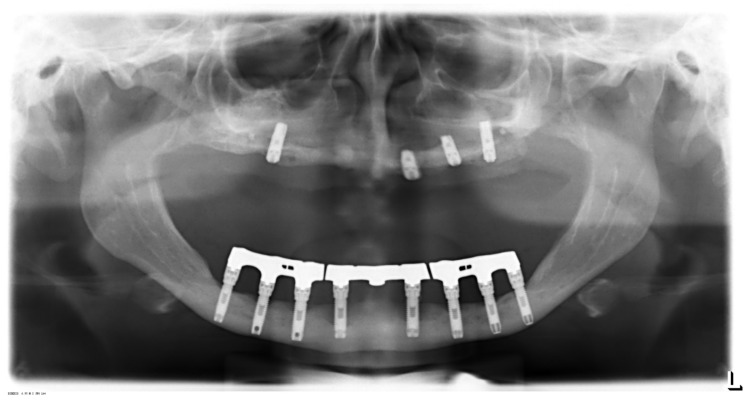
Patient 3 (panoramic radiograph performed in 2008): Eight hollow-screw implants (mandible) inserted in 1992 (16 years in situ); Four Astra^®^ implants (maxilla) inserted in 2008 (six months in situ), two implants (maxilla) have already been lost.

**Figure 2 jcm-11-02438-f002:**
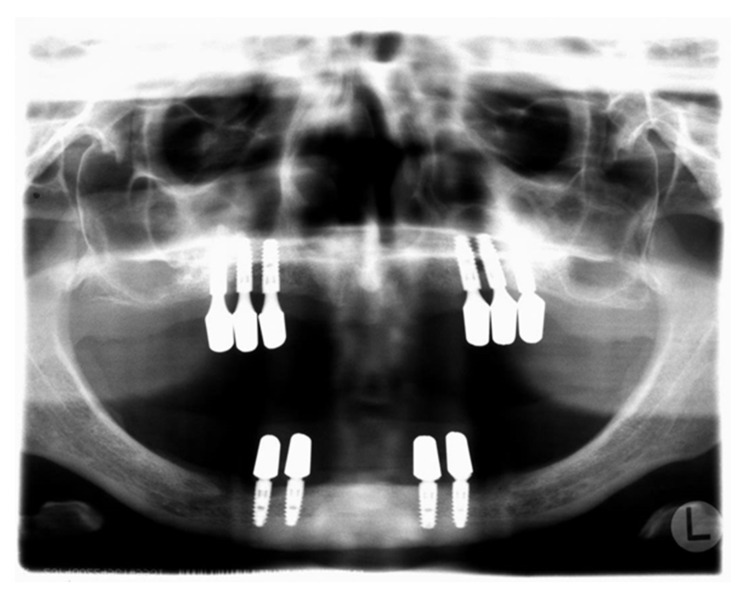
Patient 1 (panoramic radiograph performed in 2010): 10 Ankylos^®^ implants (2.5 years in situ).

**Table 1 jcm-11-02438-t001:** The patients included and their dental implant characteristics.

Family	Patient	Implants In Situ (Number/Jaw/Type); Year Implants Were Placed; Follow-Up (Years)	Implants Lost (Number)	Bone Grafting (Yes/No/Material)	Prosthetic Restoration	Supportive Therapy (Yes/No/Main Contents)
A	1 (♀/*1988)	6× maxilla 4× mandible Ankylos^®^ (Dentsply Friadent, York, PA, USA) (all Ø 3.5 mm, length 11 mm); Placed in 2007; 2.5 years	-	Yes (maxilla) Bio-Oss Block^®^ (Geistlich, Wolhusen, Switzerland), autologous bone (zygomaticum), Bio-Gide^®^ (Geistlich)	Removable telescopic crown-supported restoration (galvano)	no
A	2 (♀/*1991)	6× maxilla 4× mandible Ankylos^®^ (Dentsply Friadent, York, PA, USA) (all Ø 3.5 mm, length 9.5 and 11 mm); Placed in 2010; 5 years	-	Yes (maxilla) Bio-Oss Block^®^ (Geistlich), autologous bone (zygomaticum), Bio-Gide^®^ (Geistlich)	Removable telescopic crown-supported restoration (galvano)	yes (but irregular): professional dental cleaning once a year; measuring of PPD (irregular), subgingival cleaning (glycine) in case of increased PPD+BOP
B	3 (♀/*1974)	4× maxilla 8× mandible maxilla: Astra^®^ (Dentsply, York, PA, USA) mandible: Brånemark^®^ (Nobel Biocare, Kloten, Switzerland) Placed in 1992; re-implantation maxilla in 2008 and 2010; 20 years (mandible), 4/2 years (maxilla)	4	Yes (maxilla) Autologous bone (iliac crest)	Removable bar-carried restoration	yes: professional dental cleaning every 3 months; no measuring of PPD, no subgingival cleaning; systemic amoxicillin + clavulanic acid and metronidazole for seven days twice a year; no professional supportive therapy
B	4 (♀/*1983)	1× maxilla 4× mandible 1 disc-shaped implant (unknown manufacturer); all others: Brånemark^®^ (Nobel Biocare, Kloten, Switzerland) implants Placed in 1993; 19 years	11	Yes (maxilla) Autologous bone and bone substitute (unknown material)	Removable telescopic crown- and ball-shaped head-supported restoration	yes: professional dental cleaning every 3 months; no measurement of PPD, no subgingival cleaning; systemic amoxicillin + clavulanic acid and metronidazole for seven days twice a year; no professional supportive therapy
C	5 (♂/*1971)	5× maxilla 6× mandible 1 Biomet^®^ 3i implant, all other: IMZ^®^ implants (Dentsply, York, PA, USA); Placed in 1992; 20 years	Several implants were lost, number unknown	Yes (maxilla) (unknown material)	Removable bar-supported (maxilla) and removable telescopic crown-supported restoration (mandible)	no

♀ female, ♂ male, * born.

**Table 2 jcm-11-02438-t002:** Clinical data: probing pocket depths (PPD), bleeding on probing (BOP), suppuration, and microbiological findings at last visit.

Patient	PPD 1–3 mm (%)	PPD 4–6 mm (%)	PPD ≥ 7 mm (%)	BOP (%)	Suppuration (Yes/No)	AA +/−
1	55%	45%	0%	38%	no	AA −
2	58%	42%	0%	25%	no	AA −
3	71%	22%	7%	20%	yes	AA −
4	80%	20%	0%	9%	no	- *
5	32%	54%	14%	26%	Yes	AA −

* Patient was treated with systemic antibiotics at the time, - no microbiological examination.

**Table 3 jcm-11-02438-t003:** Bone loss around implants (bone loss in % around implant [distance implant shoulder—implant apex], mesial and distal).

Patient	Bone Loss 0− < 25%	Bone Loss 25− < 50%	Bone Loss ≥ 50%
1	75%	25%	0%
2	95%	5%	0%
3	29%	46%	25%
4	60%	20%	20%
5	13%	55%	32%

## Data Availability

The data that support the findings of this study are not available due to data protection restrictions.

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
