# Peer review of "Long-Term Results after Placing Dental Implants in Patients with Papillon-Lefèvre Syndrome: Results 2.5–20 Years after Implant Insertion"

_jcm, 2022, doi:10.3390/jcm11092438_

Round 1

Reviewer 1 Report

This study is a case series study. The number of patients is insufficient to provide the answer to the hypothesis question:  to analyze the (long-term) outcomes of dental implants in five PLS patients and to identify factors that may influence treatment outcomes. It is the biggest group of the patients, but it includes, 5 patients, and the study of Nickles et al 2011 includes 4. So this is not a significant difference to make this statement. 

In Methodology authors should explain briefly what supportive implant (SIT) therapy include.

In the results part, the authors write more on methodology regarding implant placement, and not the results of the study, and also is repetitive since most of the data is already presented in the Table. Results are not clearly presented. 

Discussion should begin and deal first with the most significant finding in the study, and it lacks here. 

Although the authors made a good question, they failed to provide an explanation. : "All patients treated with dental implants many years earlier (approximately 20 years in Patients 3, 4, and 5) exhibited advanced bone loss around the implants and suffered substantial implant loss. What could be the reasons for these failures?" Because the history of periodontal disease can not be applied to patients in this study since all of them have periodontal disease and different outcomes. 

The conclusion does not provide the answer on the aim of the study. (to analyze the (long-term) outcomes of dental implants in five PLS patients and to identify factors that may influence treatment outcomes.)

Author Response

This study is a case series study. The number of patients is insufficient to provide the answer to the hypothesis question: to analyze the (long-term) outcomes of dental implants in five PLS patients and to identify factors that may influence treatment outcomes. It is the biggest group of the patients, but it includes, 5 patients, and the study of Nickles et al 2011 includes 4. So this is not a significant difference to make this statement.

In our study from 2011 we presented data from 4 PLS patients of our actual study but in Nickles et al. 2011 implant data were presented only incidentally. Now we present data from 5 PLS patients much more detailed.

To clarify we insterted: „The authors are aware of the fact that the number of patients included into the study is very small. Anyway the manuscript clearly provides the prevalence, i.e. 1 to 4 under 1 million population. The authors are also aware of the fact that the treatment modalities (implant types, various bone grafts etc.) are very heterogeneous and hard to compare.

Inspite of everything the present study represents the largest on implant treatment in PLS patients so far.“

In Methodology authors should explain briefly what supportive implant (SIT) therapy include.

Please see Reviewer 2: To clarify we inserted: „Supportive implant therapy (SIT) consisted of oral hygiene instructions, professional dental/implant cleaning (removal of supragingiva(-mucosall plaque and dental/implant stain) and - bi-annually - a comprehensive periodontal/peri-implant charting. In this way, immediate intervention was possible whenever needed. At sites exhibiting probing pockets depths of 4 mm that showed bleeding on probing (BOP) and/or pockets that were deeper than 4 mm, subgingival debridement with titan curettes and/or air-polishing with glycine powder and instillation of a 1% chlorhexidine gel was performed. In the course of every maintenance visit, oral hygiene indices were assessed.“ to line 94-101.

In the results part, the authors write more on methodology regarding implant placement, and not the results of the study, and also is repetitive since most of the data is already presented in the Table. Results are not clearly presented.

The result section has been significantly shortened and streamlined - all data is now presented in Table 1, repetitions have been removed (line 85-107).

Discussion should begin and deal first with the most significant finding in the study, and it lacks here.

Although the authors made a good question, they failed to provide an explanation. : "All patients treated with dental implants many years earlier (approximately 20 years in Patients 3, 4, and 5) exhibited advanced bone loss around the implants and suffered substantial implant loss. What could be the reasons for these failures?" Because the history of periodontal disease can not be applied to patients in this study since all of them have periodontal disease and different outcomes.

History of periodontal disease is a risk factor for peri-implant disease in general and PLS periodontitis in particular. Thus, PLS patients are high risk patients with regard to peri-implant disease. We report 5 PLS patients losing all teeth and being treated with dental implants. Only one patient receives proper (oral hygiene indices, PPD charting, PMPR, SI) SPT on a yearly basis. Patient 1 received just one proper SPT, patient 3 and 4 received twice a year systemic antibiotics but only supramucosal PMPR, patient 5 did not receive any maintenance treatment what so ever. Thus, two main factors seem to drive bone loss: 1) time (which is trivial) and 2) lack of SPT (which is particularly evident in comparison of patients 1 and 2) (Tab. 3). We have added this to the conclusions section (please see below).

The conclusion does not provide the answer on the aim of the study. (to analyze the (long-term) outcomes of dental implants in five PLS patients and to identify factors that may influence treatment outcomes.)

We tried to clarify and inserted „History of periodontal disease is a risk factor for peri-implant disease in general and PLS periodontitis in particular. Thus, PLS patients are high risk patients with regard to peri-implant disease. We report 5 PLS patients losing all teeth and being treated with dental implants. Only one patient receives proper (oral hygiene indices, PPD charting, PMPR, SI) SPT on a yearly basis. Patient 1 received just one proper SPT, patient 3 and 4 received twice a year systemic antibiotics but only supramucosal PMPR, patient 5 did not receive any maintenance treatment what so ever. Thus, two main factors seem to drive bone loss: 1) time (which is trivial) and 2) lack of SPT (which is particularly evident in comparison of patients 1 and 2).“ to line 291-298.

Reviewer 2 Report

  1. line 38-42: the sentence is too long and hard to understand. make it simple
  2. term aggressive periodontitis: according to recent(2017) classification we cant use chronic or aggressive periodontitis
  3. line 49: Which authors? To provide more context
  4. professional dental cleaning is called scaling and root planing?
  5. line 168: provide definitions to success rate and survival rate
  6.  line 170: is this right that half the patients with healthy periodontal status are having implant failure?
  7. line 172: if only 10% had peri-implantitis what were the reason for the failure of 50% of cases in the control group of that study?
    It's important to expand on biological rationale for context even from published literature
  8. line 180: meaning of blush
  9. what constitutes a SIT?
  10. line 239: time is not mentioned so how did u presume there was no edentulous period?
  11. line 243:all these terms have been already used in the manuscript. use the abbr. terms later in the text.
  12. conclusion: 1.Mention the various modalities which could be used to maintain the implant in the oral cavity. Modalities to enhance osseo integration, preserve bone etc..
    2. write the limitations of the study

Author Response

  • line 38-42: the sentence is too long and hard to understand. make it simple
  • Has been changed accordingly: . „For an increasing number of cases of PLS patients, it is reported that periodontitis may be arrested even in the long-term. In those cases therapy consists of  treatment of the infection with extraction of severely diseased teeth, combined mechanical and antibiotic periodontal treatment, oral hygiene instructions, intensive maintenance therapy, and microbiological monitoring.“ 9, 10, 11
  • term aggressive periodontitis: according to recent (2017) classification we cant use chronic or aggressive periodontitis

The study from Swierkot et al. was made with the classification system from 1999 – sure the term „aggressive periodontitis“ is not in line with the current classification but periodontitis grade C also consists of  patients with the former diagnose „generalized severe chronic periodontitis“. We tried to clarify and changed/added to line 175-185:

The use of dental implants in patients with formerly known as aggressive periodontitis (AgP; now patients with periodontitis grade C) has already been reported. 12, 13 Patients with periodontitis grade C (AgP) had a five times greater risk of implant failure, a three times larger risk of mucositis, and a 14 times higher risk of peri-implantitis. The authors concluded that patients with treated periodontitis grade C (AgP) are more susceptible to mucositis and peri-implantitis and experience lower implant survival and success rates.” 13

GagP was changed into periodontitis grade C through the entire manuscript.

  • line 49: Which authors? To provide more context

Has been changed accordingly: Swierkot et al. was inserted.

  • professional dental cleaning is called scaling and root planing?

No, in Germany professional dental cleaning („PZR“) means removal of supragingival dental plaque and dental stain. That is why we explained in line 254 „no subgingival cleaning was performed“.

We inserted an explanation in line 253: „Although in Patients 3 and 4, professional dental cleanings (professional mechanical plaque removal (PMPR) and removal dental stain) were performed every three months, no measurement of probing depths, assessment of BOP/suppuration, or subgingival cleaning took place.“

  • line 168: provide definitions to success rate and survival rate

We insterted „The  implant  success  rate  was  defined  by  the following  parameters:  1)  no  implant  movement;  2)no discomfort (pain, foreign body sensation, paresthe-sia); 3) PD≤5 mm without BOP; 4) no continuous ra-diologic translucency; and 5) annual peri-implant boneloss≤0.2 mm 1 year after insertion of the superstruc-ture. Implants that did not meet≥1 criterion were con-sidered a failure.“                     to clarify in line 168.

  •  line 170: is this right that half the patients with healthy periodontal status are having implant failure?

We wrote: „Further, the implant success rate was 33% in GAgP patients and 50% in periodontally healthy patients.” The implant success rate was defined as above mentioned: “The  implant  success  rate  was  defined  by  the following  parameters:  1)  no  implant  movement;  2)no discomfort (pain, foreign body sensation, paresthe-sia); 3) PD≤5 mm without BOP; 4) no continuous ra-diologic

 translucency; and 5) annual peri-implant boneloss≤0.2 mm 1 year after insertion of the superstruc-ture. Implants that did not meet≥1 criterion were con-sidered a failure.“

So half of  periodontally healthy patients did not fulfill all criteria for success.

  • line 172: if only 10% had peri-implantitis what were the reason for the failure of 50% of cases in the control group of that study?

Please see above (implant success criteria).

It's important to expand on biological rationale for context even from published literature

Unfortunately it is not clear what the reviewer exactly means.

  • line 180: meaning of blush

At first sight or at first glance – „blush“ was changed accordingly.

  • what constitutes a SIT?

To clarify we inserted: „Supportive implant therapy (SIT) consited of oral hygiene instructions, professional dental cleaning (removal of supragingival plaque and dental stain) and - bi-annually - a comprehensive periodontal status. In this way, a rapid intervention was possible whenever needed. In probing pockets depths of 4 mm that showed bleeding on probing (BOP) and/or pockets that were deeper than 4 mm, subgingival scaling with titan curettes and/or air-polishing with glycine powder and instillation of a 1% chlorhexidine gel was performed. In the course of every maintenance visit, oral hygiene indices were assessed.“ to line 94-101.

  • line 239: time is not mentioned so how did u presume there was no edentulous period?

We have several x-rays from patient 3, 4 and 5: in patient 5 two remaining teeth are still in place. In patient 3, 4 and 5 it can be seen by x-rays that there was no edentuolous period for them.

  • line 243:all these terms have been already used in the manuscript. use the abbr. terms later in the text.

To clarify we deleted supportive periodontal therapy in line 259. The abbreviation for periodontally-compromised patients (PCPs)  and periodontally healthy patients (PHP) are mentioned first at this point in the manuscript.

  • conclusion: 1.Mention the various modalities which could be used to maintain the implant in the oral cavity. Modalities to enhance osseo integration, preserve bone etc..
    2. write the limitations of the study

To clarify we added „Therefore it is of crucial importance that PLS patients are informed about the importance of supportive therapy.“ to line 288/289.

To clarify we added: „The authors are aware of the fact that the number of patients included into the study is very small. Anyway the manuscript clearly provides the prevalence, i.e. 1 to 4 under 1 million population. With 82 million Germans this may be 80 to 320 PLS patients at all throughout Germany.  Inspite of everything the present study represents the largest on implant treatment in PLS patients so far.“ In line 290-293.

Reviewer 3 Report

Dear authors,

I have reviewed the article entitled ”Long-term results after placing dental implants in patients with Papillon-LefeÌ€vre syndrome –incalculable risk or real treatment 3 option?”.

I have a few remarks to do:

-even the title specify ,,long-term results”, the time factor is not taken into account. The bone loss can be related to this factor? A comparison between the bone loss related to Papillon-LefeÌ€vre syndrome and the bone loss related to time is necessary.

-the proportional bone loss was determined only by using panoramic radiographs. I consider that a more accurate determination would be by using a CBCT investigation.

-the patients are under a treatment specific for the disease? Does this treatment can enhance the bone loss? 

-I understand that the cases implant treated are a few, but it is important to take into consideration the fact that this study has a lot of variables that can interfere with the scientific conclusions/significance (implant types, various bone grafts with different resorption types, presence or absence or supportive therapy) 

-the references cited in the text does not respect the journal guidelines [exemple] 

Author Response

Dear authors,

I have reviewed the article entitled ”Long-term results after placing dental implants in patients with Papillon-LefeÌ€vre syndrome –incalculable risk or real treatment 3 option?”.

I have a few remarks to do:

-even the title specify ,,long-term results”, the time factor is not taken into account.

To clarify we changed title into „Long-term results after placing dental implants in patients with Papillon-Lefèvre syndrome – Results 2.5-20 years after implant insertion“

The bone loss can be related to this factor? A comparison between the bone loss related to Papillon-LefeÌ€vre syndrome and the bone loss related to time is necessary.

Can the reviewer 3 please explain this in more detail?

Table 1 clearly demonstrates that lack of professional supportive treatment (frequent assessment of oral hygiene and peri-implant pockets as well as submucosal cleaning) is associated with implant loss.

-the proportional bone loss was determined only by using panoramic radiographs. I consider that a more accurate determination would be by using a CBCT investigation.

Reviewer 3 is right. CBCT examination would have allowed better analysis of peri-implant bone loss. However, radiographs were obtained in clinical routine and radiation dose had to be considered. Any way, we do not have CBCTs from patient 3, 4 and 5. In patient 1 and 2 a CBCT is available but we used the lowest comparable denominator for determination of bone loss.

-the patients are under a treatment specific for the disease? Does this treatment can enhance the bone loss? 

To clarify we added: „Some PLS patients receive systemic retinoid medication. In the literature supposedly positive effects have been reported for a systemic medication with oral retinoids (Nazzaro et al. 1988, Gelmetti et al.1989), but also not for others (Lundgren &  Renvert  2004). One of our patients (patient 2) is receiving a systemic retinoid (acitretin) for several years now – despite this she has lost all of her teeth.“ to line 279-283.

-I understand that the cases implant treated are a few, but it is important to take into consideration the fact that this study has a lot of variables that can interfere with the scientific conclusions/significance (implant types, various bone grafts with different resorption types, presence or absence or supportive therapy)

To clarify we inserted „Therefore it is of crucial importance that PLS patients are informed about the importance of supportive therapy. The authors are aware of the fact that the number of patients included into the study is very small. Anyway the manuscript clearly provides the prevalence, i.e. 1 to 4 under 1 million population. The authors are also aware of the fact that the treatment modalities (implant types, various bone grafts etc.) are very heterogeneous and difficult to compare. Inspite of all limitations the present study represents the largest case series on implant treatment in PLS patients so far.“ in line 296-303.

-the references cited in the text does not respect the journal guidelines

Was changed accordingly.

Round 2

Reviewer 2 Report

the manuscript is well-drafted and all the required information are added in the revised manuscript 

Reviewer 3 Report

Dear authors,

I appreciate the efforts in respond and solve all the ossues that I have taised.

From my point of view, the article can be published!

Best regards!